# Under his thumb the effect of president Donald Trump's Twitter messages on the US stock market

Heleen Brans[1☉], Bert Scholtens[1,2☉]*

**1** Department of Finance, Faculty of Economics and Business, University of Groningen, Groningen, The Netherlands, **2** School of Management, University of Saint Andrews, St Andrews, Scotland, United Kingdom

☉ These authors contributed equally to this work.
* l.j.r.scholtens@rug.nl

**Data Availability Statement:** We retrieved daily return indexes of individual companies from Thomson Reuters DataStream, a database of global financial and macroeconomic data. We also retrieved Standard & Poor's 500 (S&P 500) daily

## Abstract

Does president Trump's use of Twitter affect financial markets? The president frequently mentions companies in his tweets and, as such, tries to gain leverage over their behavior. We analyze the effect of president Trump's Twitter messages that specifically mention a company name on its stock market returns. We find that tweets from the president which reveal strong negative sentiment are followed by reduced market value of the company mentioned, whereas supportive tweets do not render a significant effect. Our methodology does not allow us to conclude about the exact mechanism behind these findings and can only be used to investigate short-term effects.

## Introduction

> *"My daughter Ivanka has been treated so unfairly by @Nordstrom.*
>
> *She is a great person—always pushing me to do the right thing!*
>
> *Terrible!" @realDonaldTrump.*

This is a tweet from the Twitter account of the president of the United States of America, Donald J. Trump. President Trump won the elections of November 8, 2016 as the Republican candidate, and became the 45th president of the USA. The US president can have a significant influence on the American economy [1,2]. To exert this influence, one of the means is communicating with the public. In this respect, president Trump is the second US president to use Twitter to disseminate his thoughts. Although president Barack Obama also used Twitter, president Trump is the first president to extensively communicate with the public in a personal and informal manner using social media. President Trump has outspoken opinions that often lead to extensive coverage in the media.

This study investigates whether and how president Trump's tweets influence the stock market returns of mentioned companies. As such, we study the short-term impact of this method of communication on the value of these firms. President Trump is one of the most influential people on Twitter, with more than 58 million followers and some 40,000 tweets. @POTUS is

return indexes for each event from this database. All events are included in an appendix. Data is in a repository (https://hdl.handle.net/10411/VIPJIN)

**Funding:** The authors received no specific funding for this work.

**Competing interests:** The authors have no competing interests.

the official presidential Twitter account. It was previously used by president Obama and is now reserved for the exclusive use of president Trump. But the president is much more active on his own personal account, @realDonaldTrump. Therefore, our study considers only the Twitter messages posted on this account. Compared to president Obama, president Trump names publicly traded companies regularly. The @BarackObama account has no tweets naming publicly traded companies, and the @POTUS44 account shows only one tweet mentioning a publicly traded company (Lehman Brothers) during president Obama's presidential term.

Does president Trump move the markets with his tweets? To examine this question, we use an event study. A key assumption in this type of study is that the event is unexpected. We argue that this is the case with president Trump's tweets, as they relate to the president's mood and feelings about companies, which are difficult to predict. Another issue with event studies is that the information is available to market participants. As tweets can be freely accessed and, assuming market analysts monitor the Twitter account of president Trump and his comments on publicly traded companies, investors may react to the tweets as if they were public news event releases [3]. We investigate if the presidential tweets affect stock market returns in the first two years of his presidency and whether the sentiment of the tweet makes a difference.

## News, tweets, and markets

News moves stock prices [4]. News reaches the public via channels such as newspapers, television, social media, and official statements. These media channels have been studied extensively [5–8]. Social media enable president Trump to share information with the public real-time. Twitter is a social medium widely used to broadcast and share information about activities, status, and opinions. Investors show interest in Twitter too. The Securities and Exchange Commission (SEC) approved dissemination of information by companies via Twitter in 2013. Before this approval, Twitter was not considered a legitimate outlet for communication; since it violated Regulation Fair Disclosures, which relates to regulations that seek to eliminate selective disclosure. The regulation makes sure that all investors have access to the same information and no selective groups are favored. Due to the enormous amount of potentially important information that can be shared on Twitter, algorithms were developed after Twitter became a valid public disclosure source. These algorithms can detect tweets that are of interest to investors. Various studies scrutinize how social media relate to or predict stock market reactions. One study shows that frequent occurrence of financial terms in Twitter tweets is a significant predictor of daily market returns [9]. Zheludev et al. [10] find that up to 10% of all Twitter messages contain lead-time information about financial data. Further, financial information on Twitter affects the stock market [11], supporting the assumption that president Trump's tweets might influence stock market returns too. Others investigate the use of social media via "sentiment analysis", which is text analysis that systematically identifies and extracts subjective information from sources material, and argue that positively and negatively loaded tweets contain significant forecast information about the stock market index [10].

President Trump is very active on Twitter, commenting daily on recent news and events. His tweets are an important and novel source of market information to investors, since he has insider information on US government policy. The president provides high-value political information, since the markets can incorporate his tweets in the context of policy views. Another reason president Trump's tweets are interesting is that they can amount to "free advertising" for a company to a broad audience. When a company receives attention from the president, this might encourage political supporters to buy its products. As the tweets reflect the feelings of president Trump, they may not actually convey any novel information about the firm. Therefore, the value-relevance of the tweets is not self-evident and needs to be tested

in an explicit way. The stock market response helps detect whether they are financially relevant indeed.

Some studies have investigated president Trump's tweets. Afanasyev et al. [12] do so regarding the impact on the US dollar exchange rate of the Russian ruble. Other studies more closely align with our research question, namely the relationship with stock market performance. A working paper by Born et al. [13] studies 15 company specific tweets regarding ten firms in the period of his election and his swearing in ceremony. They find that both positive and negative tweets elicited abnormal returns on the event date. Juma'h and Alnsour [14] study tweets from president Trump during his campaign period and his first year of presidency. They have 58 tweets with specific company names. They combine this with tweets about immigration, employment, tax reform, finance, and the economy. They show that, on average, there are no significant effects of the tweets on market indices or share prices. Ge et al. [15] study the president's tweets in his first year of presidency and have 48 company specific tweets. They find that these tweets slightly move company stock prices, especially those before the presidential inauguration (January 20, 2017). Further, although the response to negative tweets is larger than to positive ones, the difference between the two is not statistically significant. We aim to complement this literature by substantially expanding the sample and by specifically looking into the sentiment of the tweets to find out whether positively toned tweets have less or more impact than negatively toned ones. To this extent, we use automated sentiment analysis and refrain from manual coding as in Ge et al. [15].

Our first hypothesis tests whether the president's posting of tweets that include company names has a significant influence on the stock market return of these firms. This is to find out if financial investors appreciate the information from the president's tweets as value-relevant indeed. In this respect, we want to find out if our substantially larger sample and more extended research period yield similar results as previous studies did. Then, we investigate whether the sentiment of the tweet matters. This is because, according to the finance literature, investors have more pronounced reactions to negative than to positive news [16]. Hence, our second hypothesis tests whether the stock market return after negatively toned presidential tweets is significantly negative and more pronounced than the return after positive tweets.

## Material and methods

An event study measures the impact of a specific event on the value of a firm using financial market data [17]. The usefulness of such a study results from the fact that, given rationality in the marketplace, the effects of an event will be reflected immediately in stock prices [17]. Using stock prices, it is possible to measure the economic impact of an event over a short time period [18]. The general flow of analysis in an event study is as follows (based on [17], pages 14–15): ". . . [first] define the event of interest and identify the period over which the security prices of the firms involved will be examined—the event window. [. . .] The period of interest includes the day of the announcement of the event and the day after. This captures the price effects of announcements, which occur after the market closes on the announcement day. [. . .] After identifying the event, it is necessary to determine the selection criteria for the inclusion of a firm in the event study. [. . .] The appraisal of the event's impact requires a measure of the abnormal return. [. . .] The abnormal return is the actual ex post return of the security over the event window minus the normal return of the firm over the event window. The normal return is defined as the expected return without conditioning on the event taking place. . .". As such, the event study methodology is well suited to inform about the value relevance of events, but does not allow determining the mechanisms that are behind any market response to news.

This specific event study examines tweets posted by president Trump that mention a publicly traded company. Twitter data are collected from the Twitter account of president Trump (@realDonaldTrump) over the period November 8, 2016 (Election Day) to November 8, 2018 (two years after Election Day). In this two-year period, the president posted over 5,600 Tweets. All tweets were retrieved from TrumpTwitterarchive.com/archive, this database contains all Twitter posts of President Trump. This archive contains tweets that are deleted from Trump's Twitter account as well. But only deleted tweets that were online for at least 24 hours are considered in our study. This is because we need to take into account a breakpoint in the dataset: Since 27/1/2017, the archive switched to monitor the twitter messages in real time whereas before this day the tweets were collected on a daily basis (at least once every 24 hours). This means that tweets that have been online for less than 24 hours before 27/1/2017 are missing in the database and cannot be included in the study. Therefore, we decided to remove all tweets that are deleted within this 24 hour time frame to remain consistent regarding the sampling process. To avoid 'contamination', we removed re-tweets, tweets pertaining to non-publicly listed related companies, non-specific tweets, and tweets within the event window about any other company. We also removed companies involved in merger and acquisition activity, who did a stock split, gave a profit warning, or saw a change in their top management team. The reason for doing so is that the literature usually finds a stock market response after this type of news [17]. If the president tweets about this news and we would keep this tweet in our sample, it is not possible to determine whether the market response results from the event as such or from the president's tweet. As a result, in the end, a sample of 100 tweets remains. For event studies, this is substantial and yields powerful test results [17]. However, the number is too small to allow regressing abnormal returns of tweets on presidential, firm or investor properties. This would also require a theory regarding the impact of the president's tweets on firm or investor performance, which we lack.

We retrieved daily return indexes of individual companies from DataStream, a database of global financial and macroeconomic data. We also retrieved Standard & Poor's 500 (S&P 500) daily return indexes for each event. The S&P 500 is a stock market index based on the market capitalization of 500 large companies listed on the NYSE, NASDAQ or CBOE. Due to strict requirements, the S&P 500 index is one of the most accurate indicators of the economy and stock market of the United States. Therefore, we use the S&P500 as the market index (i.e., benchmark) for our event study. To arrive at the normal (expected) stock market return, we use an estimation window of 250 days (ranging from [–251, –1]) to estimate the expected returns in the event window [17,18]. This window allows for appropriate estimation of expected returns [19]. Day 0 is the event day, and the event window includes 2 days (day 0 and day 1). We use this short event window to maximize the capturing-effect of the event on stock prices, while minimizing influence from other factors [17,20]. With a longer event window, the likelihood of a substantial impact on market returns from other (confounding) factors increases considerably. The event window starts at the announcement day, because pre-event leakage is unlikely, as the tweets contain the president's personal opinions.

Twitter operates around the clock, and therefore there are many tweets posted outside stock exchange trading hours. To ensure that the market response to a tweet is considered in relation to the closest trading time, they are separated into two groups: tweets inside trading hours and tweets outside trading hours. The first group relates to tweets occurring between 9:30 am and 4:00 pm EST. For these tweets, we use market closing prices. We acknowledge that the lack of access to high-frequency market information is a limitation of our study. For the group occurring outside trading hours, we use next-day market opening prices. S1 File (column 6), specifies all tweets as occurring either inside trading hours ('closing') or outside trading hours ('opening'). When president Trump posts a tweet in the weekend or on a

holiday, it is assigned to the next trading day, since this is when investors are able to react to the tweet. Sometimes a tweet consists of multiple tweets, due to the limited number of words that Twitter allows. These tweets can be recognized as ending with a series of dots, followed by another message within a few minutes starting with dots. We consider such events as one single event. When there are multiple tweets concerning one company assigned the same opening or closing rate, this is considered one event as well.

The market and risk adjusted returns model (market model) is the most frequently used model in event studies. This model relates the return of a security to the return of the market portfolio (in our case the S&P 500). The market model assumes a constant linear relation between the security and the S&P 500 [17]. In our event study, the securities of the companies differ between events (see S1 File) because the president twitters about different firms. The returns of the securities are calculated by dividing the logarithm of the daily price of the stock $\ln(P_{it})$ by the logarithm of the price on the previous day $\ln(P_{i,(t-1)})$. To determine the impact of the event, we calculate abnormal returns:

$$AR_{it} = R_{it} - (\alpha_i + \beta_i R_{mt})$$

where $AR_{it}$ is abnormal return, $R_{mt}$ is return of the market index, and $R_{it}$ is actual return of event $i$ on day $t$. Alpha and beta are ordinary least squares estimates of the market model. The beta coefficient shows the sensitivity between the stock market as a whole and stock $i$, whereas the alpha coefficient shows the risk involved with stock $i$. Both alpha and beta are calculated over the estimation period. From the abnormal returns, the average abnormal return (AAR) can be calculated as follows:

$$AAR_t = \frac{1}{N} \sum_{i=1}^{N} AR_{it}$$

where N is sample size, in this case the 100 Twitter messages. Aggregation of the abnormal returns yields the cumulative abnormal return (CAR):

$$CAR_i(T1, T2) = \sum_{t=T1}^{T2} AR_{it}$$

where T1 is, in this case 0, and T2 is 1, since the event window ranges from 0 to 1. From the cumulative abnormal returns, the cumulative average abnormal return (CAAR) can be calculated:

$$CAAR(T1, T2) = \frac{1}{N} \sum_{i=1}^{N} CAR_i(T1, T2)$$

We perform a parametric test to determine the significance of the results. The parametric test assumes a normally distributed sample [19]. We use the crude dependence adjustment (CDA) test, since it compensates for potential dependence of returns across events (to account for the fact that president Trump's Twitter messages might influence one another). The test statistic is given as (following [19], 3081):

$$t_{CDA} = \frac{\bar{u}_t}{s(\bar{u})}$$

where $\bar{u}_t$ is the equal weighted portfolio mean return on day t, with $\bar{u}_t$ calculated as:

$$\bar{u}_t = \frac{1}{100}\sum_{i=1}^{100} u_{it}$$

where $\bar{u}_{it}$ is the abnormal return. The standard deviation is given by:

$$s(\bar{u}) = \sqrt{\frac{1}{250}\sum_{t=-251}^{-1}(\bar{u}_t - \bar{\bar{u}})^2}$$

Where $\bar{u}$ is:

$$(\bar{u}) = \frac{1}{250}\sum_{t=-251}^{-1}(\bar{u}_t)$$

Finally, to calculate $t_{CDA}$ for CAAR, the following formula is used:

$$t_{CDA(T1,T2)} = \frac{CAAR(T1, T2)}{\sqrt{(T2 - T1)} * s(\bar{u})}$$

The descriptive statistics in Table 1 reveal that there is substantial skewness in the alfa's and that their distribution is too peaked. For this reason, we perform non-parametric tests next to the parametric ones. More specifically, we use the generalized sign test where the null hypothesis assumes no (cumulative) abnormal returns [21]. For this test, the proportion of securities that have a positive (cumulative) abnormal return under the null hypothesis of no (cumulative) abnormal return is determined. The fraction $\hat{p}$ of a given sign of (cumulative) abnormal returns expected in the estimation period is [19]:

$$\hat{p} = \frac{1}{N}\sum_{i=1}^{N}\frac{1}{M_i}\sum_{-251}^{-1}S_{it}$$

Here, $S_{it}$ is determined by the number of positive (cumulative) abnormal returns in the estimation window; $M_i$ is the number of non-missing estimation period returns for security $i$. The generalized sign test statistic ($Z$) is:

$$Z = \frac{w - N\hat{p}}{[N\hat{p}(1 - \hat{p})]^{1/2}}$$

where $w$ is number of stocks in the event window for which the (cumulative) abnormal return is positive [19]. The main advantage of this test is that skewness in returns is taken into account.

**Table 1. Descriptive statistics of abnormal returns in the estimation window.**

|  | Alpha | Beta | AAR |
|---|---|---|---|
| **Mean** | 0.0003 | 0.979 | 0.0000 |
| **Median** | 0.0002 | 1.022 | 0.0001 |
| **Standard Deviation** | 0.0009 | 0.447 | 0.0017 |
| **Kurtosis** | 2.2730 | 0.093 | -0.0004 |
| **Skewness** | 1.1890 | -0.270 | -0.0355 |
| **Minimum** | -0.0014 | -0.088 | -.00430 |
| **Maximum** | 0.0039 | 2.009 | 0.0048 |

Beta shows the relationship between market and stock returns. Alpha shows the risk involved with the stock. The AAR over the estimation window is shown in the last column.

We divide the sample into subsamples to perform an analysis regarding the sentiment of the tweets. The particular subgroups are created using SentiStrength, which extracts sentiment strength from informal English text. SentiStrength is a highly accurate sentiment analysis tool specified for short social web texts [22,23]. It is produced as a part of the CyberEmotions project which is supported by the EU FP7; This tool is able to detect social media grammar and misspellings (such as HAPPPPY, omg, :-, b-day, and what's up) [23]. SentiStrength gives all tweets a score from -5 (very strong negative emotion or energy) to +5 (very strong positive emotion or energy). SentiStrength has a word list with positive and negative sentiment terms and their strength; some examples: dislike (-3), hate (-4), excruciating (-5), lover (4), coolest (3) and encourage (2). Moreover apostrophes are taken into account in this sentiment analysis as well; depending on the context of the tweet it is given a positive notation or a negative notation. Each tweet gets a positive sentiment score between 0 and 5 and a negative sentiment score between 0 and -5 (see S1 File for an example). These sentiment scores are combined by summing up the positive and negative scores to obtain a single sentiment polarity; in this study, the polarity ranges from -4 to +3 (S1 File). These SentiStrength codes are then recoded to -1 for any negative number, to +1 for any positive number, and to zero for neutral (S1 File). We end with 37 Twitter messages with a negative sentiment and 44 with a positive sentiment (leaving 19 neutrals). SentiStrength is tested for accuracy among the tweets and can predict positive emotion with 60.6% accuracy and negative emotion with 72.8% accuracy [22]. Therefore, we checked all tweets to detect whether tweets were wrongly assigned. As a result, tweet number 79 (see S1 File) was deleted from this analysis.

We test for differences between the different groups with a two-sample t-test. To calculate the t-value, the following formula is used:

$$t = \frac{AAR_1 - AAR_2}{\sqrt{s^2 \left( \frac{1}{n_1} + \frac{1}{n_1} \right)}}$$

In this formula, $n_1$ and $n_2$ are the number of events in each subgroup. $s^2$ is determined via the following formula:

$$s^2 = \frac{\sum_{i=1}^{n1} (AR_i - AAR_1)^2 + \sum_{i=1}^{n2} (AR_i - AAR_2)^2}{n_1 + n_2 - 2}$$

The hypothesis used in the difference test is: H0: $AAR_1 = AAR_2$ versus H1: $AAR_1 \neq AAR_2$. We make the following three comparisons: negative sentiment vs. positive sentiment, to see whether a positive tweet has a different AAR than a negative tweet; negative versus all tweets, to see if tweets with negative sentiment have a different effect compared to all tweets, and the same for positive tweets.

## Presidential powers

Descriptive statistics of the sample returns are shown in Table 1. It appears that the beta is on average 0.979, which means that the stocks are only slightly less sensitive to the market (S&P500) than average. As such, this shows that our sample very well reflects the US stock market as a whole. Fig 1 shows AARs for all events in the two-day event window. The figure reveals that one event (#87) is a notable outlier, as it reflects a drop in stock price of almost 20%. The tweet associated with this event is the following: *"Twitter 'SHADOW BANNING' prominent Republicans. Not good. We will look into this discriminatory and illegal practice at once! Many complaints"*. Twitter was under fire in this period due to its fake-account purge. President Trump reacted to this news item by stating that it is not legal and will be

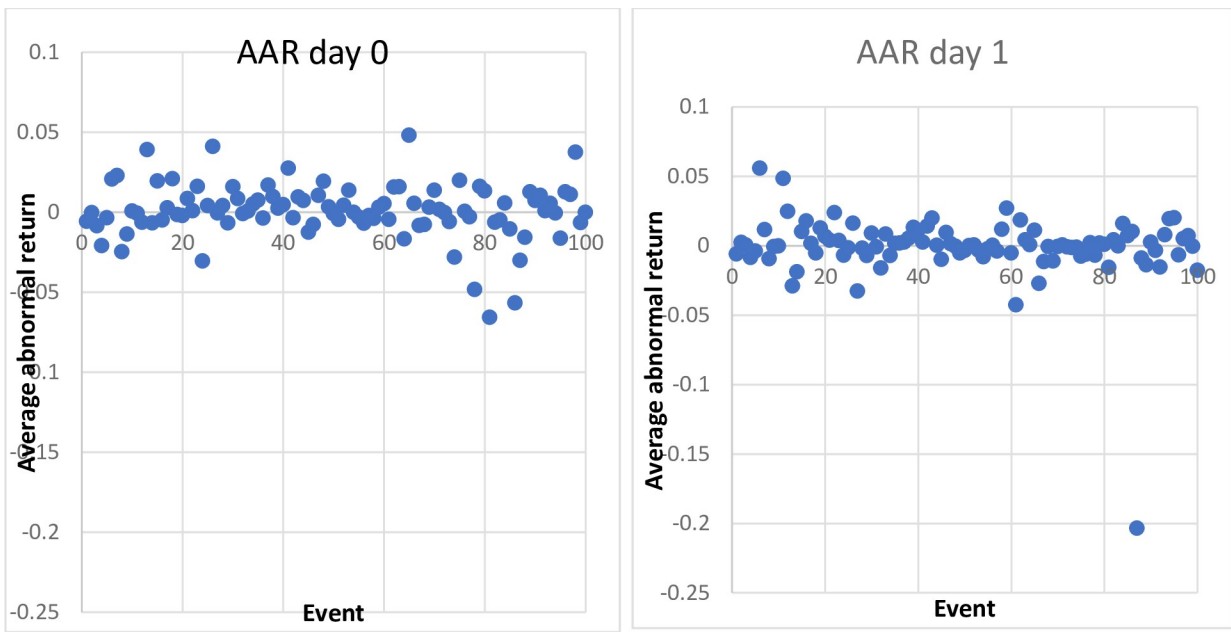

**Fig 1. Average abnormal stock market returns in percentages on days 0 and 1 for all events.**

investigated. Given its extreme magnitude, the event was removed from this sample; including it would have strengthened our results. No other outliers were detected.

Table 2 reports results from our estimations of the (cumulative) AARs in the event window. This table shows that AAR on the event day is 0.1%, which is not significant at any significance level according to both tests (i.e., parametric and non-parametric). On the first day after the event day, the AAR is -0.07%, which is not significant at any level either. Further, the CAAR of 0.5% in the event window [0,1] is not statistically significant. Therefore, the null hypothesis of no abnormal returns after a tweet from president Trump cannot be rejected. Thus, we conclude that on the day of the tweet about a company and on the first day thereafter, there is no statistically significant effect on the stock market value of the mentioned company. As such, it seems the president does not move the stock market with his tweets in a statistically significant way and the tweets are not economically meaningful. These results confirm those from Juma'h and Alnsour [14] and Ge et al. [15] for a much smaller sample. However, they contrast with those of Born et al. [13], who studied a sample of fifteen tweets between the presidential elections and the swearing in of president Trump.

To test our second hypothesis regarding the sentiment of the tweet, we investigate positive, negative, and neutral tweets. The hypothesis of no difference in AAR between subgroups on the event day is tested via a two-sided difference test. We specifically compare whether the difference between responses to negative and positive tweets is significantly different from zero.

**Table 2. Estimation results and test statistics of (cumulative) AARs in relation to president Trump's tweets with companies named.**

| Day | AAR | *p*-value parametric test | *p*-value non-parametric test |
|---|---|---|---|
| 0 | 0.0010 | 0.5383 | 0.5636 |
| 1 | -0.0007 | 0.6544 | 0.5636 |
| Window | CAAR | | |
| [0; 1] | 0.0051 | 0.5098 | 0.5402 |

This is reported in Table 3. This table shows that the AARs for the two subgroups are significantly different from zero at the 10% level for both subgroups on the event day (day 0) according to the non-parametric test. The AARs for one day after the event day are statistically significant for the negative tweets only. Further, the difference in the AARs between the two subsamples is 0.7% and 1.0%, significant at the 10% level. Therefore, the hypothesis that the AARs of the two subgroups are not different from each other can be rejected. This contrasts with the results from Ge et al. [15], who do not find an effect. The difference might be related to the fact that we study more presidential tweets over a longer period, where market participants became to realize presidential tweets might be financially relevant for companies. However, such change in their views is hard to test.

We perform additional difference tests regarding the sentiment of the tweets by using the information from the SentiStrength scores. First, we compare positive tweets and all other tweets (i.e., neutral and negative). Here, the difference between the AARs is 0.25% and 0.35%, respectively, for day 0 and day 1 (S1 File). A one-sided t-test compares the groups. This test shows that the difference between the tweets which reveal positive sentiment tweets and all other tweets is not statistically significant. Further, we compare tweets which reveal negative sentiment tweets and all others. Here, there is a marginally significant underperformance of negative tweets on the event day, but not so on the first day thereafter (S1 File). In addition, we account for the strength of the opinion expressed in the tweets. We first compare the market response to very strong opinions (SentiStrength scores -4, -3, 3, and 4) with that to moderate to neutral opinions (SentiStrength scores -1, 0, and 1). Here, it shows (S1 File) that the investor response to the strongest opinions significantly deviates from that to neutral and moderate ones on the event day. We also investigate whether the response to strong positive opinions in relation to neutral ones differs from the response to strong negative ones. Here, we use scores in the (absolute value of) 2–4 range and compare with a SentiStrength score of 0. We observe (S1 File) that there is a significant difference on very strong negative tweets on day 0. Comparing the very strong positive tweets (2, 3, 4) to the very strong negative tweets (-2,-3,-4) gives a significant difference on day 0 as well.

Thus, regarding the second hypothesis, we observe that accounting for the sentiment of the tweet informs our analysis. This especially is the case with tweets from the president which reveal strong negative sentiment.

## Discussion

We study the impact of president Trump's Twitter messages on the stock market. We investigate 100 of his tweets that include the name of a publicly listed company over the first two years of his presidency. We also carry out an analysis of the sentiment of the tweets by using

Table 3. Comparing the response to tweets with negative and positive sentiment.

| Day | 0 | 1 |
|---|---|---|
| AAR negative tweets | -0.0037 | -0.0071 |
| p-value parametric test | 0.0781 | 0.0046 |
| p-value non-parametric test | 0.0905 | 0.0905 |
| AAR positive tweets | 0.0035 | 0.0028 |
| p-value parametric test | 0.0830 | 0.1346 |
| p-value non-parametric test | 0.0523 | 0.1502 |
| Difference between positive and negative tweets | 0.0072 | 0.0098 |
| p-value of difference | 0.0808 | 0.0931 |

textual analysis. Overall, the president's tweets did not yield a significant response from the stock market. However, if we account for the sentiment of the tweets, we observe that especially tweets with a (strong) negative sentiment tweets render a significant negative response from the investment community in an economically meaningful way. This is in line with previous research [14,15] and confirms that investors are more sensitive to bad news than to good news [9,16]. We feel that our systematic sampling approach, the substantially larger number of events, the inclusion of non-parametric testing, and relying on textual analysis regarding the sentiment of the tweets, contributes to a better understanding of the economic impact of communications from president Trump.

There are limitations to this event study. First, the tweets are selected manually to find tweets with company names mentioned. This could lead to exclusion and inclusion from incorrect events in the dataset. Further, this study uses daily data; for future research, high-frequency data may be studied, since social media, especially tweets, diffuse rapidly among investors. Future research could also focus on whether the effects of the president's tweets become less strong over time as people become accustomed to his way of using Twitter. It would also be interesting to compare our results with another president or influencer.

Our results show that president Trump's tweets about companies have a negative impact on stock market returns when they are revealing strong negative sentiment. The firms themselves usually are not in a position to respond.

## Supporting information

**S1 File.**
(DOCX)

## Author Contributions

**Conceptualization:** Bert Scholtens.

**Investigation:** Heleen Brans.

**Methodology:** Heleen Brans, Bert Scholtens.

**Project administration:** Heleen Brans, Bert Scholtens.

**Supervision:** Bert Scholtens.

**Validation:** Bert Scholtens.

**Writing – original draft:** Heleen Brans.

**Writing – review & editing:** Bert Scholtens.

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
