## [Decision Letter · Decision Letter 0]

30 Oct 2019

PONE-D-19-26269

Under His Thumb The Effect of President Donald Trump’s Twitter Messages on the US Stock Market

PLOS ONE

Dear Mr Scholtens,

Thank you for submitting your manuscript to PLOS ONE. After careful consideration, we feel that it has merit but does not fully meet PLOS ONE’s publication criteria as it currently stands. Therefore, we invite you to submit a revised version of the manuscript that addresses the points raised during the review process.

In particular, we would like you to address the issues raised by the two referees about the data processing, the size of the dataset and the limitations of this study. The two referees also raised concerns about the terminology used to refer the temperament of President Trump and also asked for a more detailed explanation and justification of the methodology. We ask you to make sure that the methods are described in sufficient detail to enable reproducibility and replicability.

We would appreciate receiving your revised manuscript by Dec 14 2019 11:59PM. To enhance the reproducibility of your results, we recommend that if applicable you deposit your laboratory protocols in protocols.io, where a protocol can be assigned its own identifier (DOI) such that it can be cited independently in the future. For instructions see: http://journals.plos.org/plosone/s/submission-guidelines#loc-laboratory-protocols

We look forward to receiving your revised manuscript.

Kind regards,

Alexandre Bovet, Ph.D.

Academic Editor

PLOS ONE

Journal Requirements:

3. Please amend your manuscript to include your abstract after the title page.

Reviewers' comments:

Reviewer's Responses to Questions

**Comments to the Author**

1. Is the manuscript technically sound, and do the data support the conclusions?

Reviewer #1: Yes

Reviewer #2: Partly

2. Has the statistical analysis been performed appropriately and rigorously? 

Reviewer #1: Yes

Reviewer #2: Yes

3. Have the authors made all data underlying the findings in their manuscript fully available?

Reviewer #1: Yes

Reviewer #2: Yes

4. Is the manuscript presented in an intelligible fashion and written in standard English?

Reviewer #1: No

Reviewer #2: Yes

5. Review Comments to the Author

Reviewer #1: The authors investigate the effects of President Donald Trump's Twitter use on financial returns of companies he mentions at the daily timescale using an event study methodology. The analysis is conducted appropriately, however some clarifications are needed in order for the manuscript to be publishable in my opinion.

1) The authors declare they eliminate from their analysis any tweet that was online for less than 24 hours. This seems to me an unnecessary restriction, since financial markets react on a much faster time scale nowadays and tweet deletion doesn't necessarily imply that it would have no effect. I invite the authors to expand on the justification for this choice of threshold;

2) The authors refer numerous times to the President's "temperamental nature". While it can be understood why they call it like that, it seems inappropriate to me on a scientific journal and would suggest they find an agreement with the editor on the choice of wording;

3) I would suggest the authors introduce the event study methodology and statistical testing with more detail, giving particular attention to the distinction between quantities calculated in the estimation period or in the observation period.

4) In the definition of the CDA test statistic, \\overline{u} and \\overline{u_it} appear in the formulas but are never defined. This makes the explanation particularly confusing. I strongly suggest that the authors revise the whole section and spend more words explaining the methods, since they cannot be easily understood from the text as they are written;

5) Reference to the original paper by Cowan (Cowan, A. R. (1992). Nonparametric event study tests. Review of Quantitative Finance and Accounting, 2(4), 343-358) should be there;

6) In the definition of the non-parametric test, no clear definition is given for quantities M_i and S_it. Also, there is some confusing phrasing regarding positive abnormal returns (line 224) and negative abnormal returns (line 227), which should be made more clear.

7) In Table 1, a p-value for the Jarque-Bera test would be easier to understand for readers who don't know the quantiles of the Chi Squared distribution by heart;

8) Figure 1 panels should be made of the same size and with same x axis limits in order to be easier to read and compare. Also, axis labels instead of plot titles would make the plot easier to understand at a glance.

Reviewer #2: The paper presents an analysis of the effects of president Trump twitter messages on the stock market. The paper is rigorous in the data analysis and comply with the strict requirements of PLOS ONE with regards to data availability and statistical analysis. However, I have a number of complaints about the data used in the paper, the presentation and justification of the data analysis techniques employed and the strength and interpretation of the conclusions. I think these complaints should be acted upon to make the paper better.

I detailed my comments in the following:

1) Data

The dataset used in the analysis, in my view, is too restrictive. There is no really justification on the reasons it should be based just on the first two years of presidency of Trump. I can understand the reasons of the starting date, but there is no real reason to restrict it to just two years. In fact, it would have been much more useful to extend it to the present day (so one additional year) and then study if the activity of the president actually made the found effects on the market and perhaps showed some decay over time. The major problem is that the data set is too small, once irrelevant or questionable tweets are removed. The analysis is based on only 100 tweets! Not to mention that these 100 tweets then become only 81 once sentiment analysis is applied to them, (assuming the 19 neutral tweets did not find any use)! The set is too small, in my view, to support the analysis that followed as other factors happening at the same time could have produced the observed result. The use of more data would certainly make this possibility smaller.

For the same reasons I would also extend the event window from 2 to maybe 3 or 4 days. This would enable to account for holidays or other delaying factors on the effect the tweet could have on the market. Also, an analysis of the effects of the window size on the conclusions would also be quite interesting.

Minor comment:

- the distinction between "estimation window" and "event window" is not clear.

- why did you not had access to more high-frequency information? One of the authors is from a school of management and so this should not have been so difficult.

2) Data analysis techniques

While the formulation of the hypnotises and style of the analysis is very clear and presented with rigour, the motivations for the choice of some of the techniques has not been clearly justified. Some more explanation on the reasons you chose the "market and risk adjusted returns model" would have been useful. More importantly, why did you choose the "crude dependence adjustment model"? And could you provide some reference for it?

Why did you use the Jarque-Bera statistics? Why not one of the many other statistics? The choice of some of these methods should be clearly justified so as not to suggest the choice could have been biased by the results.

Minor comment:

- what does the index t stands for in the formula at page 8 line 208?

- why loosing the strength of the sentiment provided by SentiStrength and just accounting for its positive and negative nature (as reported in line 244, page 10)? It would have been quite interesting to relate the strength of the opinion to its effects. It could have provided also more interesting arguments for the analysis reported at the end of page 14.

- in the discussion you mention an "analysis of the tone of the tweets by using textual analysis". If this is a reference to the use you did of the results provided by SentiStrength, I am afraid it is not that strong!

3) Interpretation of the results

I fund quite strange that in a scientific paper the authors could make such "personal" and "opinionated" comments such as: "We assume that the temperamental nature of the president results in lack of predictability" (page 3, line 64) or again "temperamental nature" (page 7, line 163) or, finally, your mention of the possibility that someone with prior knowledge of the president's tweets my able able to profit from them (page 15, line 351). I think such comments are better left out from a scientific paper as to avoid possible attacks that would lower the scientific value of the conclusions.

-- Minor point:

page 4, line 106: "the may" -> "may"

6. PLOS authors have the option to publish the peer review history of their article (what does this mean?). If published, this will include your full peer review and any attached files.

Reviewer #1: No

Reviewer #2: No

---

## [Author Response · Author response to Decision Letter 0]

5 Dec 2019

Our response has also been uploaded. Format below is not as insightful as in the document uploaded.

# Comment Response 

Reviewer 1

1 The authors investigate the effects of President Donald Trump's Twitter use on financial returns of companies he mentions at the daily timescale using an event study methodology. The analysis is conducted appropriately, however some clarifications are needed in order for the manuscript to be publishable in my opinion. 

* Thank you very much for taking the time to read our manuscript and provide suggestions.

2 The authors declare they eliminate from their analysis any tweet that was online for less than 24 hours. This seems to me an unnecessary restriction, since financial markets react on a much faster time scale nowadays and tweet deletion doesn't necessarily imply that it would have no effect. I invite the authors to expand on the justification for this choice of threshold; 

* Thanks for raising this issue. The decision to delete tweets that were online for less than 24 hours is based upon a limitation on the data archive. Unfortunately, TrumpTwitterarchive.com/archive has a breakpoint in their data series, from 27/1/2017 onwards the archive switched to monitoring the twitter tweets in real time. Before this date, the tweets were only collected daily (at least every 24 hours). Therefore, not all tweets that are deleted can be collected in the data archive. With certainty, it can be said that all the tweets that have been online for over 24 hours are included in the dataset and therefore the decision has been made to eliminate all tweets that are deleted within the time span of 24 hours to avoid a breakpoint in the dataset that is unrelated to the topic studied. {161-171} 

3 The authors refer numerous times to the President's "temperamental nature". While it can be understood why they call it like that, it seems inappropriate to me on a scientific journal and would suggest they find an agreement with the editor on the choice of wording; 

* This point is well taken. We amend the choice of wording when we refer to the tweets of the US president. Reviewer 2 also raised this issue (see comment #19) {65-66, 196-197, 360}

4 I would suggest the authors introduce the event study methodology and statistical testing with more detail, giving particular attention to the distinction between quantities calculated in the estimation period or in the observation period.

* We provide more detail about the methodology and the tests employed. We point out the importance of the estimation period for the calculations of the AARs in the event period. {143-156}

5 In the definition of the CDA test statistic, \\overline{u} and \\overline{u_it} appear in the formulas but are never defined. This makes the explanation particularly confusing. I strongly suggest that the authors revise the whole section and spend more words explaining the methods, since they cannot be easily understood from the text as they are written; 

* We provide more detail about this test statistic. {245-257} Thanks for suggesting, please see the changes in relation to the previous comment (#4). 

6 Reference to the original paper by Cowan (Cowan, A. R. (1992). Nonparametric event study tests. Review of Quantitative Finance and Accounting, 2(4), 343-358) should be there; 

* Thanks for suggesting this; we include it in the list of references. {261, 432-433}

7 In the definition of the non-parametric test, no clear definition is given for quantities M_i and S_it. Also, there is some confusing phrasing regarding positive abnormal returns (line 224) and negative abnormal returns (line 227), which should be made more clear. 

* We amended the text; see also response regarding comment #5., We provide more detail. {245-257, 259-267}

8 In Table 1, a p-value for the Jarque-Bera test would be easier to understand for readers who don't know the quantiles of the Chi Squared distribution by heart. 

* We include the p-value. {307; last line in the table}

9 Figure 1 panels should be made of the same size and with same x axis limits in order to be easier to read and compare. Also, axis labels instead of plot titles would make the plot easier to understand at a glance. 

* Thanks for suggesting this; we changed the formatting and included the axis labels. {322-326}

Reviewer 2

10 The paper presents an analysis of the effects of president Trump twitter messages on the stock market. The paper is rigorous in the data analysis and comply with the strict requirements of PLOS ONE with regards to data availability and statistical analysis. However, I have a number of complaints about the data used in the paper, the presentation and justification of the data analysis techniques employed and the strength and interpretation of the conclusions. I think these complaints should be acted upon to make the paper better. 

* We very much appreciate you took the time to read our manuscript and to provide comments. 

11 The dataset used in the analysis, in my view, is too restrictive. There is no really justification on the reasons it should be based just on the first two years of presidency of Trump. I can understand the reasons of the starting date, but there is no real reason to restrict it to just two years. In fact, it would have been much more useful to extend it to the present day (so one additional year) and then study if the activity of the president actually made the found effects on the market and perhaps showed some decay over time. The major problem is that the data set is too small, once irrelevant or questionable tweets are removed. The analysis is based on only 100 tweets! Not to mention that these 100 tweets then become only 81 once sentiment analysis is applied to them, (assuming the 19 neutral tweets did not find any use)! 

* This is a comment that is highly relevant in the light of conducting an event study; The issue of how many events are ‘enough’ is heavily debated. There are a large number of event studies that rely on less than 10 events and event studies which have one event only.

First, all tweets were checked for company names; this relates to about 7000 tweets per year. Companies are not mentioned that often and we selected all of them as described in the manuscript {158-180}. Compared to existing research, it shows that the related studies have much less tweets in the sample than we do, namely Born et al. – 15; Ge et al. – 48; Juma’h and Alnsour – 58.

Second, to substantiate that our sample size is not overly restrictive, we refer to MacKinlay (1997; table 2, page 97) and the simulations by Brown and Warner (1985). This suggests that our sample size is not too small from an event study perspective. 

Last, an event study is not very suitable to investigate long-time effects of news. Most importantly, this is because of the continuous build-up of confounding events (news) that happens after the original event.

12 The set is too small, in my view, to support the analysis that followed as other factors happening at the same time could have produced the observed result. The use of more data would certainly make this possibility smaller. 

* See also our response to the previous comment. We control for confounding news regarding the companies as explained in the data section. {171-174}

13 For the same reasons I would also extend the event window from 2 to maybe 3 or 4 days. This would enable to account for holidays or other delaying factors on the effect the tweet could have on the market. Also, an analysis of the effects of the window size on the conclusions would also be quite interesting. 

* Thanks for raising this important issue. We account for holidays and weekends by using trading days throughout the analysis. We realize this might not have been very clear and now explain it in the text. {207-209} Further expanding the event window conflicts with the assumptions of relying on the market and risk adjusted returns model and would reduce the power of all test stats. The assumption is that market participants can respond quickly to news and that their combined efforts reflect in the stock prices (returns). Changing to a long event window reduces the power of the tests and opens up the impact of more and more confounding events (news). In addition, it would rely on the assumption that investors would start to respond to a tweet after some days. This is not in line with finance theory though.

14 the distinction between "estimation window" and "event window" is not clear.

* Thank you for highlighting this; Reviewer 1 also raises the point (comment #4). We now understand the need to explain the research method and provide a more detailed explanation of the event study method and point out the differences between the two windows. {143-156}

15 Why did you not had access to more high-frequency information? One of the authors is from a school of management and so this should not have been so difficult. 

* Unfortunately, our university is not endowed well and is not willing to fund access to the databases that include HF information / tick data.

16 While the formulation of the hypnotises and style of the analysis is very clear and presented with rigour, the motivations for the choice of some of the techniques has not been clearly justified. Some more explanation on the reasons you chose the "market and risk adjusted returns model" would have been useful. More importantly, why did you choose the "crude dependence adjustment model"? And could you provide some reference for it? Why did you use the Jarque-Bera statistics? Why not one of the many other statistics? The choice of some of these methods should be clearly justified so as not to suggest the choice could have been biased by the results. 

* See also the response to comments #12-14. We provide more detail about the event study methodology and motivate the use of the market and risk adjusted model, CDA, and provide reference. {143-156, 241-271, 432-433} We use the JB to inform about the (non)normality of the distribution. We do perform both parametric and non-parametric testing to rule out bias. 

17 what does the index t stands for in the formula at page 8 line 208? 

* The ‘t’ is for the day in the event window. We now also mention this in the accompanying text. {248}

18 Why loosing the strength of the sentiment provided by SentiStrength and just accounting for its positive and negative nature (as reported in line 244, page 10)? It would have been quite interesting to relate the strength of the opinion to its effects. It could have provided also more interesting arguments for the analysis reported at the end of page 14. 

* Thank you very much for suggesting this. It is very useful. We include an assessment of whether the strength of sentiment matters. We study whether (very) ‘strong’ expressions yield abnormal returns that significantly deviate from moderate/neutral. We study whether ‘strong’ positive yields a different response than ‘strong’ negative responses. {375-384; see also Supplementary Material D and E}

19 I fund quite strange that in a scientific paper the authors could make such "personal" and "opinionated" comments such as: "We assume that the temperamental nature of the president results in lack of predictability" (page 3, line 64) or again "temperamental nature" (page 7, line 163) or, finally, your mention of the possibility that someone with prior knowledge of the president's tweets my able able to profit from them (page 15, line 351). I think such comments are better left out from a scientific paper as to avoid possible attacks that would lower the scientific value of the conclusions. 

* Thanks for raising this issue (Reviewer 1 also has brought it up – comment #3). We realize that we should have toned down as it might distract from the argumentation. We amended the choice of words in the text fragments mentioned in this comment. {65-66, 196-197, 360}

We deleted the remark about prior knowledge. {415-416}

20 page 4, line 106: "the may" -> "may" 

* ‘they’ {110}

---

## [Decision Letter · Decision Letter 1]

27 Dec 2019

PONE-D-19-26269R1

Under His Thumb

The Effect of President Donald Trump’s Twitter Messages on the US Stock Market

PLOS ONE

Dear Mr Scholtens,

Thank you for submitting your manuscript to PLOS ONE. After careful consideration, we feel that it has merit but does not fully meet PLOS ONE’s publication criteria as it currently stands. Therefore, we invite you to submit a revised version of the manuscript that addresses the points raised during the review process.

The reviewers are mostly satisfied with your first revision but raised a few more points that need to be addressed before we can consider the manuscript for publication.

In addition to the points raised by the reviewers, we also ask you to better describe how the sentiment analysis (SentiStrength) works and how could one use it the reproduce the results.

We also ask you to clearly acknowledge the limitations of observational studies and causality analysis in the abstract and in the main text.

A data availability statement also needs to be added in the manuscript (see https://journals.plos.org/plosone/s/data-availability).

We would appreciate receiving your revised manuscript by Feb 10 2020 11:59PM. To enhance the reproducibility of your results, we recommend that if applicable you deposit your laboratory protocols in protocols.io, where a protocol can be assigned its own identifier (DOI) such that it can be cited independently in the future. For instructions see: http://journals.plos.org/plosone/s/submission-guidelines#loc-laboratory-protocols

We look forward to receiving your revised manuscript.

Kind regards,

Alexandre Bovet, Ph.D.

Academic Editor

PLOS ONE

Reviewers' comments:

Reviewer's Responses to Questions

**Comments to the Author**

1. If the authors have adequately addressed your comments raised in a previous round of review and you feel that this manuscript is now acceptable for publication, you may indicate that here to bypass the “Comments to the Author” section, enter your conflict of interest statement in the “Confidential to Editor” section, and submit your "Accept" recommendation.

Reviewer #1: (No Response)

Reviewer #2: All comments have been addressed

2. Is the manuscript technically sound, and do the data support the conclusions?

Reviewer #1: Partly

Reviewer #2: Yes

3. Has the statistical analysis been performed appropriately and rigorously? 

Reviewer #1: Yes

Reviewer #2: Yes

4. Have the authors made all data underlying the findings in their manuscript fully available?

Reviewer #1: Yes

Reviewer #2: Yes

5. Is the manuscript presented in an intelligible fashion and written in standard English?

Reviewer #1: Yes

Reviewer #2: Yes

6. Review Comments to the Author

Reviewer #1: I appreciate the replies and modifications the authors provided in response to the comments both by me and by the other reviewer, which I believe have made the manuscript more readable and easy to understand. I still have two comments which I think need to be addressed before the paper is sound for publication.

1) While I agree with the authors on the choice of only two days in their event window to avoid confounding events and I think 100 events are sufficient to draw some conclusions, I also think that showing some robustness to sample selection would reinforce the results, given the significance of the tests is not particularly strong. I think the authors didn't respond appropriately to the main issue presented by reviewer #2 in their comment #11 (whom I thank for raising the issue which I didn't notice at first), where it was asked a reason for the choice of limiting their sample to the first two years of presidency instead of considering tweets up to present day. Unless the authors have a sound justification for this choice, I would suggest they perform their analysis on the updated dataset.

2) At line 250 (page 10) in the revised manuscript I believe w should be the number of stocks with positive CAR, not negative, in order to be consistent with the statements above.

Reviewer #2: This is the second version of the paper and I see that all my comments have been considered and taken action upon, as far as it waspossible, given the available data. I also appreciate very much the extension of the work taking into consideration the actual numerical strength of the sentiments provided by SentiStrength. I think this add value to the paper.

I still have a few minor comments that I would like you to address before the paper could be published:

1) You introduce the use of the S&P 500 at page 4. I believe most readers will know what it is. Yet, given its importance in your analysis, a few lines of introduction could be useful to those that have poor knowledge about this index.

2) Please re-write your first hypothesis reported at the end of page 5. It is not clear.

3) I really appreciate your clear explanation of what an "event study" is, at page 6. It is really clear now and also justify the small window of time you considered.

4) You should motive more clearly some of the paper's decision, like for example the one related to removing companies involved in mergers and acquisitions (page 7). Why? The president could have also commented on them too! Also, at page 11, you (still) did not justify the use of the Jarque-Bera statistics for a normal distribution. Why this specific statistics and not, for example the Kolmogorov–Smirnov or the Lilliefors tests, which are better known?

5) At page 11, Mike Thelwall surname is misspelled at line 276. In general, please double check the text as you might have added some weird english with your editing (e.g. page 18, like 397-398)

6) Page 13, you added a line to table 1 with the p-value of the Jarque-Bera statistics. Is 0 correct? Explain it.

7) page 19, you removed a sentence on future research on the effects in time of the president tweets. I thought that was very interesting. Why removing it? Also, the very last sentence of the paper reads odd. Written in that way, in my view, it seems to say that you could not establish any lasting effects, while what you wanted to say is that you could not find any experimental support to say that there were lasting effects, although there could have been. The two things are very different and you might want to measure clearly your words when you discuss such "political" topics.

7. PLOS authors have the option to publish the peer review history of their article (what does this mean?). If published, this will include your full peer review and any attached files.

Reviewer #1: No

Reviewer #2: No

---

## [Author Response · Author response to Decision Letter 1]

4 Feb 2020

PONE-D-19-26269R1

Dear Editor,

We very much appreciate the opportunity to revise and resubmit our manuscript to PLOS ONE. We want to thank you and the reviewers for taking the time to review the manuscript and to provide thoughtful and constructive comments. We address each of them and provide detail below. The line numbers in the response to the comments refers to the document with track changes.

Thank you very much,

Sincerely,

# Comment * Response

Editor

1 Thank you for submitting your manuscript to PLOS ONE. After careful consideration, we feel that it has merit but does not fully meet PLOS ONE’s publication criteria as it currently stands. Therefore, we invite you to submit a revised version of the manuscript that addresses the points raised during the review process. 

 * Thank you very much. We address the points below and hope you feel this results in a manuscript that complies with the publication criteria.

2 In addition to the points raised by the reviewers, we also ask you to better describe how the sentiment analysis (SentiStrength) works and how could one use it the reproduce the results.

 * Thanks for raising this, we provide more detail about the sentiment analysis in the main text (lines 264-273) and in the supplementary material (S.A). We did not include this in the main text though as we felt it would break the flow of the analysis.

3 We also ask you to clearly acknowledge the limitations of observational studies and causality analysis in the abstract and in the main text.

 *We discuss the limitations of the study in the abstract (lines 23-25) and in the main text (lines 148-150).

4 A data availability statement also needs to be added in the manuscript (see https://journals.plos.org/plosone/s/data-availability).

 * We include a data availability statement (lines 454-457) and include an appendix with essential information about the events (Supplementary Material F – Event list ).

Reviewer #1

5 I appreciate the replies and modifications the authors provided in response to the comments both by me and by the other reviewer, which I believe have made the manuscript more readable and easy to understand. I still have two comments which I think need to be addressed before the paper is sound for publication. Thank you very much. * We feel your comments have been very helpful.

6 1) While I agree with the authors on the choice of only two days in their event window to avoid confounding events and I think 100 events are sufficient to draw some conclusions, I also think that showing some robustness to sample selection would reinforce the results, given the significance of the tests is not particularly strong. I think the authors didn't respond appropriately to the main issue presented by reviewer #2 in their comment #11 (whom I thank for raising the issue which I didn't notice at first), where it was asked a reason for the choice of limiting their sample to the first two years of presidency instead of considering tweets up to present day. Unless the authors have a sound justification for this choice, I would suggest they perform their analysis on the updated dataset. 

* Thanks for raising this issue. Reviewer #2 had the same question and (s)he was ok with our response (we assume that our responses to the reviewers’ comments are available to both reviewers). An important reason we cannot expand right now is the lack of funding. We have no research budget to take on the additional highly demanding investigation of the tweets. We want to pursue research on this topic and at a later stage compare the impact during several relevant subperiod of the presidency (e.g. first versus second half; first term versus second term, relate to the business cycle). But such comparison requires more observations which are not available yet. We hope that this paper also helps us position well in the competition for research funding which we require to do the additional research which proves very time intensive. 

7 2) At line 250 (page 10) in the revised manuscript I believe w should be the number of stocks with positive CAR, not negative, in order to be consistent with the statements above. 

*Thanks for pointing this out, you are correct (see also Campbell et al., 2010, page 3081). We amended the text (lines 257-258).

 

Reviewer #2

8 This is the second version of the paper and I see that all my comments have been considered and taken action upon, as far as it was possible, given the available data. I also appreciate very much the extension of the work taking into consideration the actual numerical strength of the sentiments provided by SentiStrength. I think this add value to the paper. 

I still have a few minor comments that I would like you to address before the paper could be published:

 * Thank you very much. We feel your comments have been very helpful. 

9 1) You introduce the use of the S&P 500 at page 4. I believe most readers will know what it is. Yet, given its importance in your analysis, a few lines of introduction could be useful to those that have poor knowledge about this index. 

* Thanks for highlighting this. In order to avoid confusion, we now refer to the stock market index in general on page 4 (lines 93-94). We provide more detail about the S&P500 in the (lines 193-194).

10 2) Please re-write your first hypothesis reported at the end of page 5. It is not clear. 

* We rewrote the first hypothesis (lines 125-126).

11 3) I really appreciate your clear explanation of what an "event study" is, at page 6. It is really clear now and also justify the small window of time you considered. * Thank you very much. We are pleased that you feel the event study is clearly explained and motivated.

12 4) You should motive more clearly some of the paper's decision, like for example the one related to removing companies involved in mergers and acquisitions (page 7). Why? The president could have also commented on them too! Also, at page 11, you (still) did not justify the use of the Jarque-Bera statistics for a normal distribution. Why this specific statistics and not, for example the Kolmogorov–Smirnov or the Lilliefors tests, which are better known? 

* Thanks for raising this issue. We provide an explanation as to why we control for specific confounding news (lines 168-171). 

Further, we admit that we don’t have a strong justification for using a specific test to judge the normality of the distribution. Therefore, we decided to leave out the JB-statistic and its probability value (see also comment #14 below) but instead directly refer to the skewness and kurtosis (lines 244-246; 294).

13 5) At page 11, Mike Thelwall surname is misspelled at line 276. In general, please double check the text as you might have added some weird english with your editing (e.g. page 18, like 397-398) 

* We look into this and amended accordingly (e.g., line 264). 

We are not sure what you mean by weird English on page 18 (line 397-398) as this is a reference to an article. We did not change the title of the articles in the reference list.

14 6) Page 13, you added a line to table 1 with the p-value of the Jarque-Bera statistics. Is 0 correct? Explain it. 

* Based on your remark regarding the use of tests for normality, we decided to leave out the JB stat and the probability value (see remark #12 above). 

15 7) Page 19, you removed a sentence on future research on the effects in time of the president tweets. I thought that was very interesting. Why removing it? Also, the very last sentence of the paper reads odd. Written in that way, in my view, it seems to say that you could not establish any lasting effects, while what you wanted to say is that you could not find any experimental support to say that there were lasting effects, although there could have been. The two things are very different and you might want to measure clearly your words when you discuss such "political" topics. 

* We include the future research on the effects in time (lines 392-394). We left out the last sentence (lines 399-400) as we do not want to discuss politics.

---

## [Decision Letter · Decision Letter 2]

19 Feb 2020

Under His Thumb

The Effect of President Donald Trump’s Twitter Messages on the US Stock Market

PONE-D-19-26269R2

Dear Dr. Scholtens,

We are pleased to inform you that your manuscript has been judged scientifically suitable for publication and will be formally accepted for publication once it complies with all outstanding technical requirements.

With kind regards,

Alexandre Bovet, Ph.D.

Academic Editor

PLOS ONE

Additional Editor Comments (optional):

Please make sure that the additional data (at https://hdl.handle.net/10411/VIPJIN) is openly accessible to anyone reading the article.

Reviewers' comments:

Reviewer's Responses to Questions

**Comments to the Author**

1. If the authors have adequately addressed your comments raised in a previous round of review and you feel that this manuscript is now acceptable for publication, you may indicate that here to bypass the “Comments to the Author” section, enter your conflict of interest statement in the “Confidential to Editor” section, and submit your "Accept" recommendation.

Reviewer #1: All comments have been addressed

Reviewer #2: All comments have been addressed

2. Is the manuscript technically sound, and do the data support the conclusions?

Reviewer #1: Yes

Reviewer #2: Yes

3. Has the statistical analysis been performed appropriately and rigorously? 

Reviewer #1: Yes

Reviewer #2: Yes

4. Have the authors made all data underlying the findings in their manuscript fully available?

Reviewer #1: Yes

Reviewer #2: Yes

5. Is the manuscript presented in an intelligible fashion and written in standard English?

Reviewer #1: Yes

Reviewer #2: Yes

6. Review Comments to the Author

Reviewer #1: I believe the authors properly addressed all the comments raised by both reviewers, thus I recommend to accept the manuscript for publication.

Reviewer #2: The paper has been much improved compared to the first version I received. I think the comments of the reviewers were instrumental to that. The authors has addressed all my comments and I believe the paper is now ready to be accepted and published.

7. PLOS authors have the option to publish the peer review history of their article (what does this mean?). If published, this will include your full peer review and any attached files.

Reviewer #1: No

Reviewer #2: No

---

## [Editor Report · Acceptance letter]

24 Feb 2020

PONE-D-19-26269R2 

Under His Thumb
The Effect of President Donald Trump’s Twitter Messages on the US Stock Market 

Dear Dr. Scholtens:

I am pleased to inform you that your manuscript has been deemed suitable for publication in PLOS ONE. Congratulations! Your manuscript is now with our production department. 

With kind regards,

on behalf of

Dr. Alexandre Bovet 

Academic Editor

PLOS ONE